# Random Permutation Online Isotonic Regression

**Wojciech Kotłowski**
Poznań University of Technology
Poland
wkotlowski@cs.put.poznan.pl

**Wouter M. Koolen**
Centrum Wiskunde & Informatica
Amsterdam, The Netherlands
wmkoolen@cwi.nl

**Alan Malek**
MIT
Cambridge, MA
amalek@mit.edu

## Abstract

We revisit isotonic regression on linear orders, the problem of fitting monotonic functions to best explain the data, in an online setting. It was previously shown that online isotonic regression is unlearnable in a fully adversarial model, which lead to its study in the fixed design model. Here, we instead develop the more practical random permutation model. We show that the regret is bounded above by the excess leave-one-out loss for which we develop efficient algorithms and matching lower bounds. We also analyze the class of simple and popular forward algorithms and recommend where to look for algorithms for online isotonic regression on partial orders.

## 1 Introduction

A function $f : \mathbb{R} \to \mathbb{R}$ is called *isotonic* (non-decreasing) if $x \le y$ implies $f(x) \le f(y)$. Isotonic functions model monotonic relationships between input and output variables, like those between drug dose and response [25] or lymph node condition and survival time [24]. The problem of *isotonic regression* is to find the isotonic function that best explains a given data set or population distribution. The isotonic regression problem has been extensively studied in statistics [1, 24], which resulted in efficient optimization algorithms for fitting isotonic functions to the data [7, 16] and sharp convergence rates of estimation under various model assumptions [26, 29].

In *online learning* problems, the data arrive sequentially, and the learner is tasked with predicting each subsequent data point as it arrives [6]. In *online isotonic regression*, the natural goal is to predict the incoming data points as well as the best isotonic function in hindsight. Specifically, for time steps $t = 1, \ldots, T$, the learner observes an instance $x_i \in \mathbb{R}$, makes a prediction $\widehat{y}_i$ of the true label $y_i$, which is assumed to lie in $[0, 1]$. There is no restriction that the labels or predictions be isotonic. We evaluate a prediction $\widehat{y}_i$ by its squared loss $(\widehat{y}_i - y_i)^2$. The quality of an algorithm is measured by its *regret*, $\sum_{t=1}^{T}(\widehat{y}_i - y_i)^2 - L_T^*$, where $L_T^*$ is the loss of the best isotonic function on the entire data sequence.

Isotonic regression is nonparametric: the number of parameters grows linearly with the number of data points. It is thus natural to ask whether there are efficient, provably low regret algorithms for online isotonic regression. As of yet, the picture is still very incomplete in the online setting. The first online results were obtained in the recent paper [14] which considered linearly ordered domains in the adversarial *fixed design* model, i.e. a model in which all the inputs $x_1, \ldots, x_T$ are given to the learner before the start of prediction. The authors show that, due to the nonparametric nature of the problem, many textbook online learning algorithms fail to learn at all (including Online Gradient Descent, Follow the Leader and Exponential Weights) in the sense that their worst-case regret grows linearly with the number of data points. They prove a $\Omega(T^{\frac{1}{3}})$ worst case regret lower bound, and develop a matching algorithm that achieves the optimal $\tilde{O}(T^{\frac{1}{3}})$ regret. Unfortunately, the fixed design assumption is often unrealistic. This leads us to our main question: *Can we design methods for online isotonic regression that are practical (do not hinge on fixed design)?*

**Our contributions**  Our long-term goal is to *design practical and efficient methods for online isotonic regression*, and in this work we move beyond the fixed design model and study algorithms that do not depend on future instances. Unfortunately, the completely adversarial design model (in which the instances are selected by an adaptive adversary) is impossibly hard: every learner can suffer linear regret in this model [14]. So in order to drop the fixed design assumption, we need to constrain the adversary in some other way. In this paper we consider the natural *random permutation model*, in which all $T$ instances and labels are chosen adversarially before the game begins but then are presented to the learner in a random order.

This model corresponds with the intuition that the data gathering process (which fixes the order) is independent of the underlying data generation mechanism (which fixes instances and labels). We will show that learning is possible in the random permutation model (in fact we present a reduction showing that it is not harder than adversarial fixed design) by proving an $\tilde{O}(T^{\frac{1}{3}})$ upper bound on regret for an online-to-batch conversion of the optimal fixed design algorithm from [14] (Section 3).

Our main tool for analyzing the random permutation model is the *leave-one-out loss*, drawing interesting connections with cross-validation and calibration. The leave-one-out loss on a set of $t$ labeled instances is the error of the learner predicting the $i$-th label after seeing all remaining $t-1$ labels, averaged uniformly over $i = 1, \ldots, t$. We begin by proving a general correspondence between regret and leave-one-out loss for the random permutation model in Section 2.1, which allows us to use excess leave-one-out loss as a proxy for regret. We then describe a version of online-to-batch conversion that relates the fixed design model with the random permutation model, resulting in an algorithm that attains the optimal $\tilde{O}(T^{\frac{1}{3}})$ regret.

Section 4 then turns to the computationally efficient and natural class of *forward algorithms* that use an offline optimization oracle to form their prediction. This class contains most common online isotonic regression algorithms. We then show a $O(T^{\frac{1}{2}})$ upper bound on the regret for the entire class, which improves to $O(T^{\frac{1}{3}})$ for the *well-specified* case where the data are in fact generated from an isotonic function plus i.i.d. noise (the most common model in the statistics literature).

While forward algorithms match the lower bound for the well-specified case, there is a factor $T^{\frac{1}{6}}$ gap in the random permutation case. Section 4.6 proposes a new algorithm that calls a weighted offline oracle with a large weight on the current instance. This algorithm can be efficiently computed via [16]. We prove necessary bounds on the weight.

**Related work**  Offline isotonic regression has been extensively studied in statistics starting from work by [1, 4]. Applications range across statistics, biology, medicine, psychology, etc. [24, 15, 25, 22, 17]. In statistics, isotonic regression is studied in generative models [26, 3, 29]. In machine learning, isotonic regression is used for calibrating class probability estimates [28, 21, 18, 20, 27], ROC analysis [8], training Generalized Linear Models and Single Index Models[12, 11], data cleaning [13], and ranking [19]. Fast algorithms for partial ordering are developed in [16].

In the online setting, [5] bound the minimax regret for monotone predictors under logarithmic loss and [23, 10] study online nonparametric regression in general. Efficient algorithms and worst-cases regret bounds for fixed design online isotonic regression are studied in [14]. Finally, the relation between regret and leave-one-out loss was pioneered by [9] for linear regression.

## 2   Problem Setup

Given a finite set of instances $\{x_1, \ldots, x_t\} \subset \mathbb{R}$, a function $f \colon \{x_1, \ldots, x_t\} \to [0, 1]$ is *isotonic* (non-decreasing) if $x_i \leq x_j$ implies $f(x_i) \leq f(x_j)$ for all $i, j \in \{1, \ldots, t\}$. Given a set of *labeled* instances $D = \{(x_1, y_1), \ldots, (x_t, y_t)\} \subset \mathbb{R} \times [0, 1]$, let $L^*(D)$ denote the total squared loss of the best isotonic function on $D$,

$$L^*(D) := \min_{\text{isotonic } f} \sum_{i=1}^{t} (y_i - f(x_i))^2.$$

This convex optimization problem can be solved by the celebrated *Pool Adjacent Violators Algorithm* (PAVA) in time linear in $t$ [1, 7]. The optimal solution, called the *isotonic regression function*, is piecewise constant and its value on any of its levels sets equals the average of labels within that set [24].

Online isotonic regression in the random permutation model is defined as follows. At the beginning of the game, the adversary chooses data instances $x_1 < \ldots < x_T$[1] and labels $y_1, \ldots, y_T$. A permutation $\sigma = (\sigma_1, \ldots, \sigma_T)$ of $\{1, \ldots, T\}$ is then drawn uniformly at random and used to determine the order in which the data will be revealed. In round $t$, the instance $x_{\sigma_t}$ is revealed to the learner who then predicts $\widehat{y}_{\sigma_t}$. Next, the learner observes the true label $y_{\sigma_t}$ and incurs the squared loss $(\widehat{y}_{\sigma_t} - y_{\sigma_t})^2$. For a fixed permutation $\sigma$, we use the shorthand notation $L_t^* = L^*(\{(x_{\sigma_1}, y_{\sigma_1}), \ldots, (x_{\sigma_t}, y_{\sigma_t})\})$ to denote the optimal isotonic regression loss of the first $t$ labeled instances ($L_t^*$ will clearly depend on $\sigma$, except for the case $t = T$). The goal of the learner is to minimize the *expected regret*,

$$R_T := \mathbb{E}_\sigma \left[ \sum_{t=1}^{T} (y_{\sigma_t} - \widehat{y}_{\sigma_t})^2 \right] - L_T^* = \sum_{t=1}^{T} r_t,$$

where we have decomposed the regret into its per-round increase,

$$r_t := \mathbb{E}_\sigma \left[ (y_{\sigma_t} - \widehat{y}_{\sigma_t})^2 - L_t^* + L_{t-1}^* \right], \tag{1}$$

with $L_0^* := 0$. To simplify the analysis, let us assume that the prediction strategy does not depend on the order in which the past data were revealed (which is true for all algorithms considered in this paper). Fix $t$ and define $D = \{(x_{\sigma_1}, y_{\sigma_1}), \ldots, (x_{\sigma_t}, y_{\sigma_t})\}$ to be the set of first $t$ labeled instances. Furthermore, let $D_{-i} = D \backslash \{(x_{\sigma_i}, y_{\sigma_i})\}$ denote the set $D$ with the instance from round $i$ removed. Using this notation, the expression under the expectation in (1) can be written as $(y_{\sigma_t} - \widehat{y}_{\sigma_t}(D_{-t}))^2 - L^*(D) + L^*(D_{-t})$, where we made the dependence of $\widehat{y}_{\sigma_t}$ on $D_{-t}$ explicit (and used the fact that it only depends on the set of instances, not on their order). By symmetry of the expectation over permutations with respect to the indices, we have

$$\mathbb{E}_\sigma \left[ (y_{\sigma_t} - \widehat{y}_{\sigma_t}(D_{-t}))^2 \right] = \mathbb{E}_\sigma \left[ (y_{\sigma_i} - \widehat{y}_{\sigma_i}(D_{-i}))^2 \right], \quad \text{and} \quad \mathbb{E}_\sigma \left[ L^*(D_{-t}) \right] = \mathbb{E}_\sigma \left[ L^*(D_{-i}) \right],$$

for all $i = 1, \ldots, t$. Thus, (1) can as well be rewritten as:

$$r_t = \mathbb{E}_\sigma \left[ \frac{1}{t} \sum_{i=1}^{t} \left( (y_{\sigma_i} - \widehat{y}_{\sigma_i}(D_{-i}))^2 + L^*(D_{-i}) \right) - L^*(D) \right].$$

Let us denote the expression inside the expectation by $r_t(D)$ to stress its dependence on the set of instances $D$, but not on the order in which they were revealed. If we can show that $r_t(D) \le B_t$ holds for all $t$, then its expectation has the same bound, so $R_T \le \sum_{t=1}^{T} B_t$.

## 2.1 Excess Leave-One-Out Loss and Regret

Our main tool for analyzing the random permutation model is the leave-one-out loss. In the *leave-one-out* model, there is no sequential structure. The adversary picks a data set $D = \{(x_1, y_1), \ldots, (x_t, y_t)\}$ with $x_1 < \ldots < x_t$. An index $i$ is sampled uniformly at random, the learner is given $D_{-i}$, the entire data set except $(x_i, y_i)$, and predicts $\widehat{y}_i$ (as a function of $D_{-i}$) on instance $x_i$. We call the difference between the expected loss of the learner and $L^*(D)$ the expected excess *leave-one-out loss*:

$$loo_t(D) := \frac{1}{t} \left( \left( \sum_{i=1}^{t} (y_i - \widehat{y}_i(D_{-i}))^2 \right) - L^*(D) \right). \tag{2}$$

The random permutation model has the important property that the bound on the excess leave-one-out loss of a prediction algorithm translates into a regret bound. A similar result has been shown by [9] for expected loss in the i.i.d. setting.

**Lemma 2.1.** $r_t(D) \le loo_t(D)$ *for any $t$ and any data set $D = \{(x_1, y_1), \ldots, (x_t, y_t)\}$.*

*Proof.* As $x_1 < \ldots < x_t$, let $(y_1^*, \ldots, y_t^*) = \operatorname{argmin}_{f_1 \le \ldots \le f_t} \sum_{i=1}^{t} (y_i - f_i)^2$ be the isotonic regression function on $D$. From the definition of $L^*$, we can see that $L^*(D) = \sum_{i=1}^{t} (y_i^* - y_i)^2 \ge L^*(D_{-i}) + (y_i - y_i^*)^2$. Thus, the regret increase $r_t(D)$ is bounded by

$$r_t(D) = \sum_{i=1}^{t} \frac{(y_i - \widehat{y}_i)^2 + L^*(D_{-i})}{t} - L^*(D) \le \sum_{i=1}^{t} \frac{(y_i - \widehat{y}_i)^2 - (y_i - y_i^*)^2}{t} = loo_t(D). \quad \square$$

However, we note that lower bounds for $\ell oo_t(D)$ do not imply lower bounds on regret.

In what follows, our strategy will be to derive bounds $\ell oo_t(D) \leq B_t$ for various algorithms, from which the regret bound $R_T \leq \sum_{t=1}^{T} B_t$ can be immediately obtained. From now on, we abbreviate $\ell oo_t(D)$ to $\ell oo_t$, (as $D$ is clear from the context); we will also consistently assume $x_1 < \ldots < x_t$.

## 2.2 Noise free case

As a warm-up, we analyze the noise-free case (when the labels themselves are isotonic) and demonstrate that analyzing $\ell oo_t$ easily results in an optimal bound for this setting.

**Proposition 2.2.** *Assume that the labels satisfy $y_1 \leq y_2 \leq \ldots \leq y_t$. The prediction $\widehat{y}_i$ that is the linear interpolation between adjacent labels $\widehat{y}_i = \frac{1}{2}(y_{i-1} + y_{i+1})$, has*

$$\ell oo_t \leq \frac{1}{2t}, \text{ and thus } R_T \leq \frac{1}{2}\log(T+1).$$

*Proof.* For $\delta_i := y_i - y_{i-1}$, it is easy to check that $\ell oo_t = \frac{1}{4t}\sum_{i=1}^{t}(\delta_{i+1} - \delta_i)^2$ because the $L^*(D)$ term is zero. This expression is a convex function of $\delta_1, \ldots, \delta_{t+1}$. Note that $\delta_i \geq 0$ for each $i = 1, \ldots, t+1$, and $\sum_{i=1}^{t+1} \delta_i = 1$. Since the maximum of a convex function is at the boundary of the feasible region, the maximizer is given by $\delta_i = 1$ for some $i \in \{1, \ldots, t+1\}$, and $\delta_j = 0$ for all $j \in \{1, \ldots, t+1\}, j \neq i$. This implies that $\ell oo_t \leq (2t)^{-1}$. □

## 2.3 General Lower Bound

In [14], a general lower bound was derived showing that the regret of any online isotonic regression procedure is at least $\Omega(T^{\frac{1}{3}})$ for the adversarial setup (when labels and the index order were chosen adversarially). This lower bound applies regardless of the order of outcomes, and hence it is also a lower bound for the random permutation model. This bound translates into $\ell oo_t = \Omega(t^{-2/3})$.

# 3 Online-to-batch for fixed design

Here, we describe an online-to-batch conversion that relates the adversarial fixed design model with the random permutation model considered in this paper. In the fixed design model with time horizon $T_{\mathrm{fd}}$ the learner is given the points $x_1, \ldots, x_{T_{\mathrm{fd}}}$ in advance (which is not the case in the random permutation model), but the adversary chooses the order $\sigma$ in which the labels are revealed (as opposed to $\sigma$ being drawn at random). We can think of an algorithm for fixed design as a prediction function

$$\widehat{y}^{\mathrm{fd}}\big(x_{\sigma_t}|y_{\sigma_1}, \ldots, y_{\sigma_{t-1}}, \{x_1, \ldots, x_{T_{\mathrm{fd}}}\}\big),$$

for any order $\sigma$, any set $\{x_1, \ldots, x_{T_{\mathrm{fd}}}\}$ (and hence any time horizon $T_{\mathrm{fd}}$), and any time $t$. This notation is quite heavy, but makes it explicit that the learner, while predicting at point $x_{\sigma_t}$, knows the previously revealed labels and the whole set of instances.

In the random permutation model, at trial $t$, the learner only knows the previously revealed $t-1$ labeled instances and predicts on the new instance. Without loss of generality, denote the past instances by $D_{-i} = \{(x_1, y_1), \ldots, (x_{i-1}, y_{i-1}), (x_{i+1}, y_{i+1}), \ldots (x_t, y_t)\}$, and the new instance by $x_i$, for some $i \in \{1, \ldots, t\}$. Given an algorithm for fixed design $\widehat{y}^{\mathrm{fd}}$, we construct a prediction $\widehat{y}_t = \widehat{y}_t(D_{-i}, x_i)$ of the algorithm in the random permutation model. The reduction goes through an online-to-batch conversion. Specifically, at trial $t$, given past labeled instances $D_{-i}$, and a new point $x_i$, the learner plays the expectation of the prediction of the fixed design algorithm with time horizon $T^{\mathrm{fd}} = t$ and points $\{x_1, \ldots, x_t\}$ under a uniformly random time from the past $j \in \{1, \ldots, t\}$ and a random permutation $\sigma$ on $\{1, \ldots, t\}$, with $\sigma_t = i$, i.e.[2]

$$\widehat{y}_t = \mathbb{E}_{\{\sigma:\sigma_t=i\}}\left[\frac{1}{t}\sum_{j=1}^{t} \widehat{y}^{\mathrm{fd}}(x_i|y_{\sigma_1}, \ldots, y_{\sigma_{j-1}}, \{x_1, \ldots, x_t\})\right]. \tag{3}$$

Note that this is a valid construction, as the right hand side only depends on $D_{-i}$ and $x_i$, which are known to the learner in a random permutation model at round $t$. We prove (in Appendix A) that the excess leave-one-out loss of $\widehat{y}$ at trial $t$ is upper bounded by the expected regret (over all permutations) of $\widehat{y}^{\mathrm{fd}}$ in trials $1, \ldots, t$ divided by $t$:

**Theorem 3.1.** *Let $D = \{(x_1, y_1), \ldots, (x_t, y_t)\}$ be a set of $t$ labeled instances. Fix any algorithm $\widehat{y}^{\mathrm{fd}}$ for online adversarial isotonic regression with fixed design, and let $\mathrm{Reg}_t(\widehat{y}^{\mathrm{fd}} \mid \sigma)$ denote its regret on $D$ when the labels are revealed in order $\sigma$. The random permutation learner $\widehat{y}$ from (3) ensures $\ell oo_t(D) \leq \frac{1}{t} \mathbb{E}_\sigma[\mathrm{Reg}_t(\widehat{y}^{\mathrm{fd}} \mid \sigma)]$.*

This constructions allows immediate transport of the $\tilde{O}(T_{\mathrm{fd}}^{\frac{1}{3}})$ fixed design regret result from [14].

**Theorem 3.2.** *There is an algorithm for the random-permutation model with excess leave-one-out loss $\ell oo_t = \tilde{O}(t^{-\frac{2}{3}})$ and hence expected regret $R_T \leq \sum_t \tilde{O}(t^{-\frac{2}{3}}) = \tilde{O}(T^{\frac{1}{3}})$.*

# 4 Forward Algorithms

For clarity of presentation, we use vector notation in this section: $\boldsymbol{y} = (y_1, \ldots, y_t)$ is the label vector, $\boldsymbol{y}^* = (y_1^*, \ldots, y_t^*)$ is the isotonic regression function, and $\boldsymbol{y}_{-i} = (y_1, \ldots, y_{i-1}, y_{i+1}, \ldots, y_t)$ is $\boldsymbol{y}$ with $i$-th label removed. Moreover, keeping in mind that $x_1 < \ldots < x_t$, we can drop $x_i$'s entirely from the notation and refer to an instance $x_i$ simply by its index $i$.

Given labels $\boldsymbol{y}_{-i}$ and some index $i$ to predict on, we want a good prediction for $y_i$. Follow the Leader (FL) algorithms, which predict using the best isotonic function on the data seen so far, are not directly applicable to online isotonic regression: the best isotonic function is only defined at the observed data instances and can be arbitrary (up to monotonicity constraint) otherwise. Instead, we analyze a simple and natural class of algorithms which we dub *forward algorithms*[3]. We define a forward algorithm, or FA, to be any algorithm that *estimates* a label $y_i' \in [0, 1]$ (possibly dependent on $i$ and $\boldsymbol{y}_{-i}$), and plays with the FL strategy on the sequence of past data *including* the new instance with the estimated label, i.e. performs offline isotonic regression on $\boldsymbol{y}'$,

$$\widehat{\boldsymbol{y}} = \operatorname*{argmin}_{f_1 \leq \ldots \leq f_t} \left\{ \sum_{j=1}^t (y_j' - f_j)^2 \right\}, \qquad \text{where } \boldsymbol{y}' = (y_1, \ldots, y_{i-1}, y_i', y_{i+1}, \ldots, y_t).$$

Then, FA predicts with $\widehat{y}_i$, the value at index $i$ of the offline function of the augmented data. Note that if the estimate turned out to be correct ($y_i' = y_i$), the forward algorithm would suffer no additional loss for that round.

Forward algorithms are practically important: we will show that many popular algorithms can be cast as FA with a particular estimate. FA automatically inherit any computational advances for offline isotonic regression; in particular, they scale efficiently to partially ordered data [16]. To our best knowledge, we are first to give bounds on the performance of these algorithms in the online setting.

**Alternative formulation** We can describe a FA using a *minimax* representation of the isotonic regression [see, e.g., 24]: the optimal isotonic regression $\boldsymbol{y}^*$ satisfies

$$y_i^* = \min_{r \geq i} \max_{\ell \leq i} \overline{y}_{\ell, r} = \max_{\ell \leq i} \min_{r \geq i} \overline{y}_{\ell, r}, \tag{4}$$

where $\overline{y}_{\ell, r} = \frac{\sum_{j=\ell}^r y_j}{r - \ell + 1}$. The "saddle point" $(\ell_i, r_i)$ for which $y_i^* = \overline{y}_{\ell_i, r_i}$, specifies the boundaries of the *level set* $\{j : y_j^* = y_i^*\}$ of the isotonic regression function that contains $i$.

It follows from (4) that isotonic regression is monotonic with respect to labels: for any two label sequences $\boldsymbol{y}$ and $\boldsymbol{z}$ such that $y_i \leq z_i$ for all $i$, we have $y_i^* \leq z_i^*$ for all $i$. Thus, if we denote the predictions for label estimates $y_i' = 0$ and $y_i' = 1$ by $\widehat{y}_i^0$ and $\widehat{y}_i^1$, respectively, the monotonicity implies that any FA has $\widehat{y}_i^0 \leq \widehat{y}_i \leq \widehat{y}_i^1$. Conversely, using the continuity of isotonic regression $\boldsymbol{y}^*$ as a function of $\boldsymbol{y}$, (which follows from (4)), we can show that for any prediction $\widehat{y}_i$ with $\widehat{y}_i^0 \leq \widehat{y}_i \leq \widehat{y}_i^1$, there exists an estimate $y_t' \in [0, 1]$ that could generate this prediction. Hence, we can equivalently interpret FA as an algorithm which in each trial predicts with some $\widehat{y}_i$ in the range $[\widehat{y}_i^0, \widehat{y}_i^1]$.

## 4.1 Instances

With the above equivalence between forward algorithms and algorithms that predict in $[\widehat{y}_i^0, \widehat{y}_i^1]$, we can show that many of the well know isotonic regression algorithms are forward algorithms and thereby add weight to our next section where we prove regret bounds for the entire class.

**Isotonic regression with interpolation (IR-Int)[28]** Given $\boldsymbol{y}_{-i}$ and index $i$, the algorithm first computes $\boldsymbol{f}^*$, the isotonic regression of $\boldsymbol{y}_{-i}$, and then predicts with $\widehat{y}_i^{\text{int}} = \frac{1}{2}\left(f_{i-1}^* + f_{i+1}^*\right)$, where we used $f_0^* = 0$ and $f_{t+1}^* = 1$. To see that this is a FA, note that if we use estimate $y_i' = \widehat{y}_i^{\text{int}}$, the isotonic regression of $\boldsymbol{y}' = (y_1, \ldots, y_{i-1}, y_i', y_{i+1}, \ldots, y_t)$ is $\widehat{\boldsymbol{y}} = (f_1^*, \ldots, f_{i-1}^*, y_i', f_{i+1}^*, \ldots, f_t^*)$. This is because: i) $\widehat{\boldsymbol{y}}$ is isotonic by construction; ii) $\boldsymbol{f}^*$ has the smallest squared error loss for $\boldsymbol{y}_{-t}$ among isotonic functions; and iii) the loss of $\widehat{\boldsymbol{y}}$ on point $y_i'$ is zero, and the loss of $\widehat{\boldsymbol{y}}$ on all other points is equal to the loss of $\boldsymbol{f}^*$.

**Direct combination of $\widehat{y}_i^0$ and $\widehat{y}_i^1$.** It is clear from Section 4, that any algorithm that predicts $\widehat{y}_i = \lambda_i \widehat{y}_i^0 + (1 - \lambda_i)\widehat{y}_i^1$ for some $\lambda_i \in [0,1]$ is a FA. The weight $\lambda_i$ can be set to a constant (e.g., $\lambda_i = 1/2$), or can be chosen depending on $\widehat{y}_i^1$ and $\widehat{y}_i^0$. Such algorithms were considered by [27]:

$$\text{log-IVAP}: \quad \widehat{y}_i^{\log} = \frac{\widehat{y}_i^1}{\widehat{y}_i^1 + 1 - \widehat{y}_i^0}, \qquad \text{Brier-IVAP}: \quad \widehat{y}_i^{\text{Brier}} = \frac{1 + (\widehat{y}_i^0)^2 - (1 - \widehat{y}_i^1)^2}{2}.$$

It is straightforward to show that both algorithms satisfy $\widehat{y}_i^0 \le \widehat{y}_i \le \widehat{y}_i^1$ and are thus instances of FA.

**Last-step minimax (LSM).** LSM plays the minimax strategy with one round remaining,

$$\widehat{y}_i = \operatorname*{argmin}_{\widehat{y} \in [0,1]} \max_{y_i \in [0,1]} \left\{(\widehat{y} - y_i)^2 - L^*(\boldsymbol{y})\right\},$$

where $L^*(\boldsymbol{y})$ is the isotonic regression loss on $\boldsymbol{y}$. Define $L_b^* = L^*(y_1, \ldots, y_{i-1}, b, y_{i+1}, \ldots, y_t)$ for $b \in \{0, 1\}$, i.e. $L_b^*$ is the loss of isotonic regression function with label estimate $y_i' = b$. In Appendix B we show that $\widehat{y}_i = \frac{1 + L_0^* - L_1^*}{2}$ and it is also an instance of FA.

## 4.2 Bounding the leave-one-out loss

We now give a $O(\sqrt{\frac{\log t}{t}})$ bound on the leave-one-out loss for forward algorithms. Interestingly, the bound holds no matter what label estimate the algorithm uses. The proof relies on the stability of isotonic regression with respect to a change of a single label. While the bound looks suboptimal in light of Section 2.3, we will argue in Section 4.5 that the bound is actually tight (up to a logarithmic factor) for one FA and experimentally verify that all other mentioned forward algorithms also have a tight lower bound of that form for the same sequence of outcomes.

We will bound $\ell oo_t$ by defining $\delta_i = \widehat{y}_i - y_i^*$ and using the following simple inequality:

$$\ell oo_t = \frac{1}{t}\sum_{i=1}^t \left((\widehat{y}_i - y_i)^2 - (y_i^* - y_i)^2\right) = \frac{1}{t}\sum_{i=1}^t (\widehat{y}_i - y_i^*)(\widehat{y}_i + y_i^* - 2y_i) \le \frac{2}{t}\sum_{i=1}^t |\delta_i|.$$

**Theorem 4.1.** *Any forward algorithm has $\ell oo_t = O\left(\sqrt{\frac{\log t}{t}}\right)$.*

*Proof.* Fix some forward algorithm. For any $i$, let $\{j: y_j^* = y_i^*\} = \{\ell_i, \ldots, r_i\}$, for some $\ell_i \le i \le r_i$, be the level set of isotonic regression at level $y_i^*$. We need the stronger version of the minimax representation, shown in Appendix C:

$$y_i^* = \min_{r \ge i} \overline{y}_{\ell_i, r} = \max_{\ell \le i} \overline{y}_{\ell, r_i}. \tag{5}$$

We partition the points $\{1, \ldots, t\}$ into $K$ consecutive segments: $S_k = \left\{i: y_i^* \in \left[\frac{k-1}{K}, \frac{k}{K}\right)\right\}$ for $k = 1, \ldots, K-1$ and $S_K = \left\{i: y_i^* \ge \frac{K-1}{K}\right\}$. Due to monotonicity of $\boldsymbol{y}^*$, $S_k$ are subsets of the form $\{\ell_k, \ldots, r_k\}$ (where we use $r_k = \ell_k - 1$ if $S_k$ is empty). From the definition, every level set of $\boldsymbol{y}^*$ is contained in $S_k$ for some $k$, and each $\ell_k$ ($r_k$) is a left-end (right-end) of some level set.

Now, choose some index $i$, and let $S_k$ be such that $i \in S_k$. Let $y_i'$ be the estimate of the FA, and let $\boldsymbol{y}' = (y_1, \ldots, y_{i-1}, y_i', y_{i+1}, \ldots, y_t)$. The minimax representation (4) and definition of FA imply

$$
\begin{aligned}
\widehat{y}_i \;=\; \max_{\ell \leq i} \min_{r \geq i} \overline{y}'_{\ell, r} \;&\geq\; \min_{r \geq i} \overline{y}'_{\ell_k, r} \;=\; \min_{r \geq i} \left\{ \overline{y}_{\ell_k, r} + \frac{y_i' - y_i}{r - \ell_k + 1} \right\} \\
&\geq\; \min_{r \geq i} \overline{y}_{\ell_k, r} + \min_{r \geq i} \frac{y_i' - y_i}{r - \ell_k + 1} \;\geq\; \min_{r \geq \ell_k} \overline{y}_{\ell_k, r} + \min_{r \geq i} \frac{y_i' - y_i}{r - \ell_k + 1} \\
&\overset{\text{by (5)}}{\geq}\; y_{\ell_k}^* + \min_{r \geq i} \frac{-1}{r - \ell_k + 1} \;\geq\; y_{\ell_k}^* - \frac{1}{i - \ell_k + 1} \;\geq\; y_i^* - \frac{1}{K} - \frac{1}{i - \ell_k + 1}.
\end{aligned}
$$

A symmetric argument gives $\widehat{y}_i \leq y_i^* + \frac{1}{K} + \frac{1}{r_k - i + 1}$. Hence, we can bound $|\delta_i| = |\widehat{y}_i - y_i^*| \leq \frac{1}{K} + \max\left\{ \frac{1}{i - \ell_k + 1}, \frac{1}{r_k - i + 1} \right\}$. Summing over $i \in S_k$ yields $\sum_{i \in S_k} |\delta_i| \leq \frac{|S_k|}{K} + 2\left(1 + \log |S_k|\right)$, which allows the bound

$$
\ell oo_t \leq \frac{2}{t} \sum_i |\delta_i| \leq \frac{2}{K} + 4 \frac{K}{t} (1 + \log t).
$$

The theorem follows from setting $K = \Theta(\sqrt{t / \log t})$. $\qquad\square$

## 4.3 Forward algorithms for the well-specified case

While the $\ell oo_t$ upper bound of the previous section yields a regret bound $R_T \leq \sum_t O(\sqrt{\log t / t}) = \tilde{O}(T^{\frac{1}{2}})$ that is a factor $O(T^{\frac{1}{6}})$ gap from the lower bound in Section 2.3, there are two pieces of good news. First, forward algorithms do get the optimal rate in the *well-specified* setting, popular in the classical statistics literature, where the labels are generated i.i.d. such that $\mathbb{E}[y_i] = \mu_i$ with isotonic $\mu_1 \leq \ldots \leq \mu_t$.[4] Second, there is a $\Omega(t^{-\frac{1}{2}})$ lower bound for forward algorithms as proven in the next section. Together, these results imply that the random permutation model in indeed harder than the well-specified case: forward algorithms are sufficient for the latter but not the former.

**Theorem 4.2.** *For data generated from the well-specified setting (monotonic means with i.i.d. noise), any FA has $\ell oo_t = \tilde{O}(t^{-\frac{2}{3}})$, which translates to a $\tilde{O}(T^{\frac{1}{3}})$ bound on the regret.*

The proof is given in Appendix D. Curiously, the proof makes use of the existence of the seemingly unrelated optimal algorithm with $\tilde{O}(t^{-\frac{2}{3}})$ excess leave-one-out loss from Theorem 3.2.

## 4.4 Entropic loss

We now abandon the squared loss for a moment and analyze how a FA performs when the loss function is the *entropic loss*, defined as $-y \log \widehat{y} - (1 - y) \log(1 - \widehat{y})$ for $y \in [0, 1]$. Entropic loss (precisely: its binary-label version known as log-loss) is extensively used in the isotonic regression context for maximum likelihood estimation [14] or for probability calibration [28, 21, 27]. A surprising fact in isotonic regression is that minimizing entropic loss[5] leads to exactly the same optimal solution as in the squared loss case, the isotonic regression function $\boldsymbol{y}^*$ [24].

Not every FA is appropriate for entropic loss, as recklessly choosing the label estimate might result in an infinite loss in just a single trial (as noted by [27]). Indeed, consider a sequence of outcomes with $y_1 = 0$ and $y_i = 1$ for $i > 1$. While predicting on index $i = 1$, choosing $y_1' = 1$ results in $\widehat{y}_1 = 1$, for which the entropic loss is infinite (as $y_1 = 0$). Does there exists a FA which achieves a meaningful bound on $\ell oo_t$ in the entropic loss setup?

We answer this question in the affirmative, showing that the log-IVAP predictor FA gets the same excess-leave-one-out loss bound as given in Theorem 4.1. As the reduction from the regret to leave-one-out loss (Lemma 2.1) does not use any properties of the loss function, this immediately implies a bound on the expected regret. Interestingly, the proof (given in Appendix G) uses as an intermediate step the bound on $|\delta_i|$ for the *worst possible* forward algorithm which always produces the estimate being the opposite of the actual label.

**Theorem 4.3.** *The log-IVAP algorithm has $\ell oo_t = O\left(\sqrt{\frac{\log t}{t}}\right)$ for the entropic loss.*

### 4.5 Lower bound

The last result of this section is that FA can be made to have $\ell oo_t = \Omega(t^{-\frac{1}{2}})$. We show this by means of a counterexample. Assume $t = n^2$ for some integer $n > 0$ and let the labels be binary, $y_i \in \{0, 1\}$. We split the set $\{1, \ldots, t\}$ into $n$ consecutive segments, each of size $n = \sqrt{t}$. The proportion of ones ($y_i = 1$) in the $k$-th segment is equal to $\frac{k}{n}$, but within each segment all ones *precede* all zeros. For instance, when $t = 25$, the label sequence is:

$$\underbrace{10000}_{1/5} \ \underbrace{11000}_{2/5} \ \underbrace{11100}_{3/5} \ \underbrace{11110}_{4/5} \ \underbrace{11111}_{5/5},$$

One can use the minimax formulation (4) to verify that the segments will correspond to the level sets of the isotonic regression and that $y_i^* = \frac{k}{n}$ for any $i$ in the $k$-th segment. This sequence is hard:

**Lemma 4.4.** *The IR-Int algorithm run on the sequence described above has $\ell oo_t = \Omega(t^{-\frac{1}{2}})$.*

We prove the lower bound for IR-Int, since the presentation (in Appendix E) is clearest. Empirical simulations showing that the other forward algorithms also suffer this regret are in Appendix F.

### 4.6 Towards optimal forward algorithms

An attractive feature of forward algorithms is that they generalize to partial orders, for which efficient offline optimization algorithms exist. However, in Section 4 we saw that FAs only give a $\tilde{O}(t^{-\frac{1}{2}})$ rate, while in Section 3 we saw that $\tilde{O}(t^{-\frac{2}{3}})$ is possible (with an algorithm that is not known to scale to partial orders). Is there any hope of an algorithm that both generalizes and has the optimal rate?

In this section, we propose the *Heavy-$\gamma$* algorithm, a slight modification of the forward algorithm that plugs in label estimate $y_i' = \gamma \in [0, 1]$ *with weight $c$* (with unit weight on all other points), then plays the value of the isotonic regression function. Implementation is straightforward for offline isotonic regression algorithms that permit the specification of weights (such as [16]). Otherwise, we might simulate such weighting by plugging in $c$ copies of the estimated label $\gamma$ at location $x_i$.

What label estimate $\gamma$ and weight $c$ should we use? We show that the choice of $\gamma$ is not very sensitive, but it is crucial to tune the weight to $c = \Theta(t^{\frac{1}{3}})$. Lemmas H.1 and H.2 show that higher and lower $c$ are necessarily sub-optimal for $\ell oo_t$. This leaves only one choice for $c$, for which we believe

**Conjecture 4.5.** *Heavy-$\gamma$ with weight $c = \Theta(t^{\frac{1}{3}})$ has $\ell oo_t = \tilde{O}(t^{-\frac{2}{3}})$.*

We cannot yet prove this conjecture, although numerical experiments strongly suggest it. We do not believe that picking a constant label $\gamma$ is special. For example, we might alternatively predict with the average of the predictions of Heavy-1 and Heavy-0. Yet not any label estimate works. In particular, if we estimate the label that would be predicted by IR-Int (see 4.1) and the discussion below it), and we plug that in with any weight $c \geq 0$, then the isotonic regression function will still have that same label estimate as its value. This means that the $\Omega(t^{-\frac{1}{2}})$ lower bound of Section 4.5 applies.

## 5 Conclusion

We revisit the problem of online isotonic regression and argue that we need a new perspective to design practical algorithms. We study the random permutation model as a novel way to bypass the stringent fixed design requirement of previous work. Our main tool in the design and analysis of algorithms is the leave-one-out loss, which bounds the expected regret from above. We start by observing that the adversary from the adversarial fixed design setting also provides a lower bound here. We then show that this lower bound can be matched by applying online-to-batch conversion to the optimal algorithm for fixed design. Next we provide an online analysis of the natural, popular and practical class of Forward Algorithms, which are defined in terms of an offline optimization oracle. We show that Forward algorithms achieve a decent regret rate in all cases, and match the optimal rate in special cases. We conclude by sketching the class of practical Heavy algorithms and conjecture that a specific parameter setting might guarantee the correct regret rate.

**Open problem**  The next major challenge is the design and analysis of efficient algorithms for online isotonic regression on arbitrary partial orders. Heavy-$\gamma$ is our current best candidate. We pose deciding if it in fact even guarantees $\tilde{O}(T^{\frac{1}{3}})$ regret on linear orders as an open problem.

**Acknowledgments**

Wojciech Kotłowski acknowledges support from the Polish National Science Centre (grant no. 2016/22/E/ST6/00299). Wouter Koolen acknowledges support from the Netherlands Organization for Scientific Research (NWO) under Veni grant 639.021.439. This work was done in part while Koolen was visiting the Simons Institute for the Theory of Computing.

## Footnotes

[1] We assume all points $x_t$ are distinct as it will significantly simplify the presentation. All results in this paper are also valid for the case $x_1 \le \ldots \le x_T$.

[2]Choosing the prediction as an expectation is elegant but inefficient. However, the proof indicates that we might as well sample a single $j$ and a single random permutation $\sigma$ to form the prediction and the reduction would also work in expectation.

[3]The name highlights resemblance to the Forward algorithm introduced by [2] for exponential family models.

[4]The $\Omega(T^{1/3})$ regret lower bound in [14] uses a mixture of well-specified distributions and still applies.

[5]In fact, this statement applies to any Bregman divergence [24].

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
