[Supplementary Material]

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

# A    Proof of Theorem 3.1

*Proof.* Denote $X = \{x_1, \ldots, x_t\}$. By Jensen

$$\sum_{i=1}^{t} \left(y_i - \widehat{y}_i(D_{-i})\right)^2 \leq \sum_{i=1}^{t} \mathbb{E}_{\{\sigma \,:\, \sigma_t = i\}}\left[\frac{1}{t}\sum_{j=1}^{t}\left(y_i - \widehat{y}^{\mathrm{fd}}\left(x_i\big|y_{\sigma_1}, \ldots, y_{\sigma_{j-1}}, X\right)\right)^2\right]$$

$$= \sum_{j=1}^{t}\frac{1}{t}\sum_{i=1}^{t}\mathbb{E}_{\{\sigma \,:\, \sigma_t = i\}}\left[\left(y_i - \widehat{y}^{\mathrm{fd}}\left(x_i\big|y_{\sigma_1}, \ldots, y_{\sigma_{j-1}}, X\right)\right)^2\right]$$

$$= \sum_{j=1}^{t}\frac{1}{t}\sum_{i=1}^{t}\mathbb{E}_{\{\sigma \,:\, \sigma_j = i\}}\left[\left(y_t - \widehat{y}^{\mathrm{fd}}\left(x_i\big|y_{\sigma_1}, \ldots, y_{\sigma_{j-1}}, X\right)\right)^2\right]$$

$$= \sum_{j=1}^{t}\mathbb{E}_{\sigma}\left[\left(y_{\sigma_j} - \widehat{y}^{\mathrm{fd}}\left(x_{\sigma_j}\big|y_{\sigma_1}, \ldots, y_{\sigma_{j-1}}, X\right)\right)^2\right]$$

$$= \mathbb{E}_{\sigma}\left[\sum_{j=1}^{t}\left(y_{\sigma_j} - \widehat{y}^{\mathrm{fd}}\left(x_{\sigma_j}\big|y_{\sigma_1}, \ldots, y_{\sigma_{j-1}}, X\right)\right)^2\right]$$

Subtracting $L_t^* = L^*(D)$ and dividing by $t$, we find

$$\ell oo_t(D) \ \leq \ \frac{1}{t}\mathbb{E}_{\sigma}[\mathrm{Reg}_t(\widehat{y}^{\mathrm{fd}} \,|\, \sigma)]. \qquad \square$$

# B    Some facts about the last-step minimax algorithm

The last-step minimax algorithm minimizes

$$\widehat{y}_i = \operatorname*{argmin}_{\widehat{y}\in[0,1]}\ \max_{y_i\in[0,1]}\ \left\{(\widehat{y} - y_i)^2 - L^*(\boldsymbol{y})\right\},$$

where $L^*(\boldsymbol{y})$ denotes the total loss of isotonic regression on $\boldsymbol{y}$. We now argue that the function inside $\max\{\cdot\}$ is convex in $y_i$. Let us rewrite this function as:

$$(\widehat{y} - y_i)^2 - L^*(\boldsymbol{y}) \ = \ \widehat{y}^2 - 2\widehat{y}y_i + y_i^2 - \min_{\widehat{\boldsymbol{y}}}\left\{\sum_{j=1}^{t}\left(\widehat{y}_j^2 - 2\widehat{y}_j y_j + y_j^2\right)\right\}$$

$$= \ -2\widehat{y}y_i - \min_{\widehat{\boldsymbol{y}}}\left\{\sum_{j=1}^{t}\left(\widehat{y}_j^2 - 2\widehat{y}_j y_j\right)\right\} + \mathrm{const},$$

where the last term denotes expression which does not depend on $y_i$. Now, the function inside $\min\{\cdot\}$ is concave (linear) in $y_i$, hence the minimum over $\widehat{\boldsymbol{y}}$ is also concave (in $y_i$), and so the whole function is convex in $y_i$. Therefore, the maximum over $y_i$ is attained in $\{0, 1\}$.

Define $L_b^* = L^*(y_1, \ldots, y_{i-1}, b, y_{i+1}, \ldots, y_t)$ for $b \in \{0, 1\}$, i.e. $L_b^*$ is the value of isotonic regression function with label estimate $y_i' = b$. Then,

$$\widehat{y}_i \ = \ \operatorname*{argmin}_{\widehat{y}\in[0,1]}\ \max_{y_i\in\{0,1\}}\ \left\{(\widehat{y} - y_i)^2 - L_{y_i}^*\right\}.$$

We have:

$$\min_{\widehat{y}\in[0,1]}\ \max_{y_i\in\{0,1\}}\ \left\{(\widehat{y} - y_i)^2 - L_{y_i}^*\right\}$$

$$= \ \min_{\widehat{y}\in[0,1]}\ \max_{q\in[0,1]}\ q\left((\widehat{y} - 1)^2 - L_1^*\right) + (1 - q)\left((\widehat{y} - 0)^2 - L_0^*\right),$$

$$\overset{(a)}{=} \ \max_{q\in[0,1]}\ \min_{\widehat{y}\in[0,1]}\ q\left((\widehat{y} - 1)^2 - L_1^*\right) + (1 - q)\left(\widehat{y}^2 - L_0^*\right)$$

$$\overset{(b)}{=} \ \max_{q\in[0,1]}\ q(1 - q) - L_0^* + q(L_0^* - L_1^*)$$

$$\overset{(c)}{=} \ \frac{1}{4}(1 + L_0^* - L_1^*)^2 - L_0^*,$$

where $(a)$ is from the Sion's minimax theorem, $(b)$ is from plugging in the minimizer $\widehat{y} = q$, and $(c)$ is from plugging in the maximizer $q = \frac{1 + L_0^* - L_1^*}{2}$. Thus, the minimax problem has a unique saddle point $(\widehat{y}, q)$ given by $\widehat{y} = q = \frac{1 + L_0^* - L_1^*}{2}$.

To show that LSM is a forward algorithm, denote the isotonic regression on the sequence $\boldsymbol{y}^1 = (y_1, \ldots, y_{i-1}, 1, y_{i+1}, \ldots, y_t)$ as $\widehat{\boldsymbol{y}}^1$, and the isotonic regression on the sequence $\boldsymbol{y}^0 = (y_1, \ldots, y_{i-1}, 0, y_{i+1}, \ldots, y_t)$ as $\widehat{\boldsymbol{y}}^0$ (so that the loss of $\widehat{\boldsymbol{y}}^b$ is $L_b^*$). First note that from the definition of $\widehat{\boldsymbol{y}}^0$, we have

$$L_0^* = (\widehat{y}_i^0)^2 + \sum_{j \neq i} (\widehat{y}_j^0 - y_j)^2 \leq (\widehat{y}_i^1)^2 + \sum_{j \neq i} (\widehat{y}_j^1 - y_j)^2,$$

so that

$$(\widehat{y}_i^1)^2 - L_0^* \geq -\sum_{j \neq i} (\widehat{y}_j^1 - y_j)^2.$$

Furthermore, $(\widehat{y}_i^1 - 1)^2 - L_1^* = -\sum_{j \neq i} (\widehat{y}_j^1 - y_j)^2$ because of canceling of terms $(\widehat{y}_i^1 - 1)^2$. Thus,

$$\max_{y_i \in \{0,1\}} \left\{ (\widehat{y}_i^1 - y_i)^2 - L_{y_i}^* \right\} = \max_{y_i \in \{0,1\}} \left\{ -\sum_{j \neq i} (\widehat{y}_j^1 - y_j)^2, (\widehat{y}_i^1)^2 - L_0^* \right\} = (\widehat{y}_i^1)^2 - L_0^*.$$

This means that for any $\widehat{y} > \widehat{y}_i^1$,

$$\max_{y_i \in \{0,1\}} \left\{ (\widehat{y}_i^1 - y_i)^2 - L_{y_i}^* \right\} = (\widehat{y}_i^1)^2 - L_0^* < (\widehat{y})^2 - L_0^* \leq \max_{y_i \in \{0,1\}} \left\{ (\widehat{y} - y_i)^2 - L_{y_i}^* \right\}.$$

This proves that the LSM prediction must satisfy $\widehat{y}_i \leq \widehat{y}_i^1$. One can show in a similar way that also $\widehat{y}_i \geq \widehat{y}_i^0$, which implies that LSM is an instance of FA.

## C   Proof of Equation 5

Fix $i$ and let $\{j : y_j^* = y_i^*\}$ be the level set of isotonic regression at level $y_i^*$, which is a segment of the form $\{\ell_i, \ldots, r_i\}$ for some $\ell_i \leq i \leq r_i$. We will show that

$$y_i^* = \min_{r \geq i} \overline{y}_{\ell_i, r} = \max_{\ell \leq i} \overline{y}_{\ell, r_i}.$$

We will only prove the first equality, while the second can be shown analogously. Assume the contrary, that the first equality does not hold, i.e. that $y_i^* \neq \min_{r \geq i} \overline{y}_{\ell_i, r}$. First note that from the minimax representation (4), $y_i^* \geq \min_{r \geq i} \overline{y}_{\ell_i, r}$, which, given the assumption, implies that the inequality is sharp and $y_i^* > \min_{r \geq i} \overline{y}_{\ell_i, r}$. In other words, there exists $r' \geq i$ such that $\overline{y}_{\ell_i, r'} < y_i^*$. We will show that this contradicts the optimality of $\boldsymbol{y}^*$. Indeed, since $\ell_i$ is the left-end of a level set of $\boldsymbol{y}^*$, there exists sufficiently small $\delta > 0$, such that subtracting $\delta$ from all $y_j^*$ in the range $j \in \{\ell_i, \ldots, r'\}$ will not violate the isotonic constraints. At the same time, taking derivative of $\sum_{\ell_i \leq j \leq r'} (y_j^* - \delta - y_j)^2$ with respect to $\delta$ gives

$$\sum_{\ell_i \leq j \leq r'} 2(y_j - y_j^* + \delta) \leq \sum_{\ell_i \leq j \leq r'} 2(y_j - y_i^* + \delta)$$
$$= 2(r' - \ell_i + 1)(\overline{y}_{\ell_i, r'} - y_i^* + \delta)$$
$$< 0,$$

for sufficiently small $\delta$, where the first inequality is from $y_j^* \geq y_i^*$ for all $j \geq \ell_i$ (because $y_i^* = y_{\ell_i}^*$, as $i$ in in the level set $(\ell_i, r_i)$ of $\boldsymbol{y}^*$, and due to the fact that $\boldsymbol{y}^*$ is isotonic), while the second inequality is from assumption $\overline{y}_{\ell_i, r'} < y_i^*$. But this means that the loss of $\boldsymbol{y}^*$ can be improved, which contradicts the optimality of $\boldsymbol{y}^*$.

## D   Proof of Theorem 4.2

We remind that it is assumed that the labels are generated i.i.d. such that $\mathbb{E}[y_i] = \mu_i$ with isotonic means $\mu_1 \leq \ldots \leq \mu_t$.

We proceed with bounding:

$$\begin{aligned}
(\widehat{y}_i - y_i)^2 &= (\widehat{y}_i - \mu_i + \mu_i - y_i)^2 \\
&= (\widehat{y}_i - \mu_i)^2 + 2(\widehat{y}_i - \mu_i)(\mu_i - y_i) + (\mu_i - y_i)^2 \\
&= (\widehat{y}_i - y_i^* + y_i^* - \mu_i)^2 + 2(\widehat{y}_i - \mu_i)(\mu_i - y_i) + (\mu_i - y_i)^2 \\
&\le 2(\widehat{y}_i - y_i^*)^2 + 2(y_i^* - \mu_i)^2 + 2(\widehat{y}_i - \mu_i)(\mu_i - y_i) + (\mu_i - y_i)^2, \quad (6)
\end{aligned}$$

where the last inequality is from $(a+b)^2 \le 2a^2 + 2b^2$. Since $y_i^*$ is the squared Euclidean distance projection of $y_i$ onto the convex set of isotonic functions, the Pythagorean inequality holds [24]: for any isotonic function $\boldsymbol{f} = (f_1, \ldots, f_t)$,

$$\sum_i (f_i - y_i)^2 \ge \sum_i (f_i - y_i^*)^2 + \sum_i (y_i^* - y_i)^2.$$

Summing (6) over trials and subtracting the loss of isotonic regression, dividing by $t$, and then applying the Pythagorean Theorem with $\boldsymbol{f} = (\mu_1, \ldots, \mu_t)$ gives:

$$\begin{aligned}
loo_t &= \frac{1}{t}\sum_i (\widehat{y}_i - y_i)^2 - (y_i^* - y_i)^2 \\
&\le \underbrace{\frac{1}{t}\sum_i 2(\widehat{y}_i - y_i^*)^2}_{=A} + \frac{1}{t}\sum_i \underbrace{2(\widehat{y}_i - \mu_i)(\mu_i - y_i)}_{=B_i} + 3\underbrace{\frac{1}{t}\sum_i \left((\mu_i - y_i)^2 - (y_i^* - y_i)^2\right)}_{=C}.
\end{aligned}$$

The $B_i$ term disappears in expectation for each $i$: by the definition of $\mu_i$,

$$\mathbb{E}\left[(\widehat{y}_i - \mu_i)(\mu_i - y_i)\right] = (\widehat{y}_i - \mu_i)(\mu_i - \mathbb{E}[y_i]) = 0.$$

The $C$ term can be bounded by noting that for any $i$, no predictor can have a smaller expected loss than $\mu_i$, which is the minimizer of the expected loss by definition. As shown in Theorem 3.2, there exists predictor with $\tilde{O}(t^{-\frac{2}{3}})$ excess leave-one-out loss. Let $\widehat{\boldsymbol{y}}^{\mathrm{opt}}$ be any such predictor. Then

$$\frac{1}{t}\mathbb{E}\left[\sum_i (\mu_i - y_i)^2 - (y_i^* - y_i)^2\right] \le \frac{1}{t}\mathbb{E}\left[\sum_i (\widehat{y}_i^{\mathrm{opt}} - y_i)^2 - (y_i^* - y_i)^2\right] = \tilde{O}(t^{-\frac{2}{3}}).$$

Finally, the $A$ term is equal to $\frac{1}{t}\sum_i \delta_i^2$, where $\delta_i = \widehat{y}_i - y_i^*$, and can be bound using the result obtained in the proof of Theorem 4.2. We remind that in that proof, the points $\{1, \ldots, t\}$ were partitioned into $K$ consecutive segments of the form $S_k = \left\{i : y_i^* \in \left[\frac{k-1}{K}, \frac{k}{K}\right)\right\} = \{\ell_k, \ldots, r_k\}$. For any index $i$ and $S_k$ such that $i \in S_k$, we obtained the following bound on $|\delta_i|$:

$$|\delta_i| \le \frac{1}{K} + \max\left\{\frac{1}{i - \ell_k + 1}, \frac{1}{r_k - i + 1}\right\}.$$

Combining this with $(a+b)^2 \le 2a^2 + 2b^2$ and $\max\{a, b\} \le a + b$, we have that

$$\delta^2 \le \frac{2}{K^2} + \frac{2}{(i - \ell_k + 1)^2} + \frac{2}{(r_k - i + 1)^2}.$$

Summing over $i \in S_k = \{\ell_k, \ldots, r_k\}$, we conclude that

$$\begin{aligned}
\sum_{i \in S_k} (\delta_i)^2 &\le \frac{2|S_k|}{K^2} + \sum_{i=\ell_k}^{r_k}\left(\frac{2}{(i - \ell_k + 1)^2} + \frac{2}{(r_k - i + 1)^2}\right) \\
&= \frac{2|S_k|}{K^2} + 4\sum_{i=1}^{r_k - \ell_k + 1}\frac{1}{i^2} \le \frac{2|S_k|}{K^2} + \frac{2\pi^2}{3},
\end{aligned}$$

because $\sum_{i=1}^{m}\frac{1}{i^2} \le \sum_{i=1}^{\infty}\frac{1}{i^2} = \frac{\pi^2}{6}$. Summing over segments $S_1, \ldots, S_K$ and dividing by $t$ finally yields

$$\frac{1}{t}\sum_{i=1}^{t}\delta_i^2 \le \frac{2}{K^2} + \frac{2K\pi^2}{3t},$$

which produces the claim upon setting $K = \Theta(t^{\frac{1}{3}})$.

# E   Proof of Lemma 4.4

We remind the reader that the label sequence $\boldsymbol{y}$ is constructed as follow. Assume $t = n^2$ for some integer $n > 0$. We split the set $\{1, \ldots, t\}$ into $n$ consecutive segments, each of size $n = \sqrt{t}$. The labels in the $k$-th segment are chosen so that the first $k$ labels are set to 1, while the remaining $n - k$ labels are set to 0. More formally, if $\{m + 1, \ldots, m + n\}$, for $m = (k-1)n$, are the indices in the $k$-th segment, and $y_{m+1}, \ldots, y_{m+n}$ are the corresponding labels, then $y_{m+1} = \ldots = y_{m+k} = 1$ and $y_{m+k+1} = \ldots = y_{m+n} = 0$. The proportion of ones in the $k$-th segment is thus equal to $\frac{k}{n}$. Since all ones precede all zeros in each segment, one can use the minimax formulation (4) to verify that the segments will correspond to the level sets of the isotonic regression and that $y_i^* = \frac{k}{n}$ for any $i$ in the $k$-th segment.

Take the $k$-th segment with $k$ ones preceding $n - k$ zeros. For simplicity, denote the starting index of the $k$-th segment by 1, and the final index by $n$. IR-Int works by performing isotonic regression on all except the $i$-th label and then predicting with a linear interpolation of the two adjacent points $i - 1$ and $i + 1$. Since the proportions in both adjacent segments $((k-1)$-th and $(k+1)$-th) are separated by $\frac{1}{n}$ from the proportion in the $k$-the segment, and the removal of a single label from the $k$-th segment affects its proportion by *less* than $\frac{1}{n}$, this removal only affects the value of isotonic regression *locally*, i.e., only in the $k$-th segment (this can be verified using the minimax formulation (4)).

The proportion in the $k$-th segment is equal to $\frac{k-1}{n-1}$ if one of the first $k$ labels was removed, and $\frac{k}{n-1}$ otherwise. Since $\widehat{y}_i$ is the interpolation of $y_{i-1}^*$ and $y_{i+1}^*$, it follows that $\widehat{y}_i = \frac{k-1}{n-1}$ for $2 \le i \le k$, and $\widehat{y}_i = \frac{k}{n-1}$ for $k+1 \le i \le n-1$. For the boundary points $i \in \{1, n\}$ it is a bit more complicated as we interpolate with the end points of adjacent segments, but it is enough to note that $\widehat{y}_1 \le \frac{k-1}{n-1}$ and $\widehat{y}_n \ge \frac{k}{n-1}$. The contribution to the excess leave-one-out loss of the $k$-th segment is

$$\sum_{i=1}^{n} (\widehat{y}_i - y_i)^2 - (y_i^* - y_i)^2 \ge k \left( \left( \frac{n-k}{n-1} \right)^2 - \left( \frac{n-k}{n} \right)^2 \right) + (n-k) \left( \left( \frac{k}{n-1} \right)^2 - \left( \frac{k}{n} \right)^2 \right)$$

$$= \frac{k(n-k)(2n-1)}{n(n-1)^2} \ge \frac{2k(n-k)}{n^2}.$$

Summing over the segments gives

$$loo_t \ge \frac{1}{n^2} \sum_{k=1}^{n} \frac{2k(n-k)}{n^2} = \frac{2}{n^4} \sum_{k=1}^{n} k(n-k) = \frac{2}{n^4} \left( \frac{n^2(n+1)}{2} - \frac{n(n+1)(2n+1)}{6} \right)$$

$$= \frac{1}{3n} - \frac{1}{3n^3} = \frac{1}{3\sqrt{t}} - \frac{1}{3t\sqrt{t}} = \Omega \left( \frac{1}{\sqrt{t}} \right).$$

# F   Empirical simulations of forward algorithm lower bound

We take the following seven variants of FA: IR-Int, log-IVAP, Brier-IVAP, Alg-1 $(\widehat{y}_i = \widehat{y}_i^1)$, Alg-0 $(\widehat{y}_i = \widehat{y}_i^0)$, Alg-$\frac{1}{2}$ $(\widehat{y}_i = \frac{1}{2}\widehat{y}_i^0 + \frac{1}{2}\widehat{y}_i^1)$, and LSM. Below, we present the excess leave-one-out loss of each algorithm on a log-log plot as a function of $t$ starting from $t = 2^{10} = 1\,024$ and increasing up to $t = 2^{16} = 65\,536$. See Figure 1.

The values of $loo_t \sqrt{t}$ for each algorithm are nearly flat and close to each other, at a level of around 0.337, very close to the analytically calculated lower bound of $\frac{1}{3} - \frac{1}{3t}$ for IR-Int.

# G   Proof of Theorem 4.3

Let $\ell(y, \widehat{y}) = -y \log \widehat{y} - (1 - y) \log(1 - \widehat{y})$ denote the entropic loss. The excess leave-one-out loss $loo_t$ is then given by

$$loo_t = \frac{1}{t} \sum_{i=1}^{t} \left( \ell(y_i, \widehat{y}_i) - \ell(y_i, y_i^*) \right).$$

The log-IVAP predictor is defined as

$$\widehat{y}_i = \frac{\widehat{y}_i^1}{\widehat{y}_i^1 + 1 - \widehat{y}_i^0},$$

where $\widehat{y}_i^1$ (respectively $\widehat{y}_i^0$) is the prediction at index $i$ of isotonic regression on the sequence $(y_1, \ldots, y_{i-1}, 0, y_{i+1}, \ldots, y_t)$ (respectively $(y_1, \ldots, y_{i-1}, 1, y_{i+1}, \ldots, y_t)$). We have

$$
\begin{aligned}
\ell(y_i, \widehat{y}_i) - \ell(y_i, y_i^*) &= y_i \log \frac{y_i^*}{\widehat{y}_i} + (1 - y_i) \log \frac{1 - y_i^*}{1 - \widehat{y}_i} \\
&\leq \frac{y_i}{\widehat{y}_i}(y_i^* - \widehat{y}_i) + \frac{1 - y_i}{1 - \widehat{y}_i}\left((1 - y_i^*) - (1 - \widehat{y}_i)\right) \\
&= \frac{y_i y_i^*}{\widehat{y}_i} + \frac{(1 - y_i)(1 - y_i^*)}{1 - \widehat{y}_i} - 1, \quad\quad\quad (7)
\end{aligned}
$$

where the first inequality follows from the bound $\log \frac{a}{b} = \log\left(1 + \frac{a-b}{b}\right) \leq \frac{a-b}{b}$. Using the definition of log-IVAP,

$$\frac{y_i^*}{\widehat{y}_i} = \frac{(\widehat{y}_i^1 + 1 - \widehat{y}_i^0)y_i^*}{\widehat{y}_i^1} = y_i^* + (1 - \widehat{y}_i^0)\frac{y_i^*}{\widehat{y}_i^1} \leq y_i^* - \widehat{y}_i^0 + 1,$$

where we used $y_i^* \leq \widehat{y}_i^1$, which follows from the monotonicity of isotonic regression with respect to the labels. Similarly:

$$\frac{1 - y_i^*}{1 - \widehat{y}_i} = \frac{(\widehat{y}_i^1 + 1 - \widehat{y}_i^0)(1 - y_i^*)}{1 - \widehat{y}_i^0} = 1 - y_i^* + \widehat{y}_i^1 \frac{1 - y_i^*}{1 - \widehat{y}_i^0} \leq \widehat{y}_i^1 - y_i^* + 1,$$

where we again used monotonicity argument to bound $y_i^* \geq \widehat{y}_i^0$. Plugging these bounds into (7) gives:

$$\ell(y_i, \widehat{y}_i) - \ell(y_i, y_i^*) \leq y_i(y_i^* - \widehat{y}_i^0) + (1 - y_i)(\widehat{y}_i^1 - y_i^*) \leq \max_{b \in \{0,1\}} |\widehat{y}_i^b - y_i^*|.$$

Since $\widehat{y}_i^0$ and $\widehat{y}_i^1$ are predictions of forward algorithms (with label estimates $y_i' = 0$ and $y_i' = 1$, respectively) we can directly bound the maximum on the right-hand side using the proof of Theorem 4.1. Specifically, we partition points $\{1, \ldots, t\}$ into $K$ consecutive segments of the form $S_k = \left\{i \colon y_i^* \in \left[\frac{k-1}{K}, \frac{k}{K}\right]\right\} = \{\ell_k, \ldots, r_k\}$. For any index $i$ and $S_k$ such that $i \in S_k$, we obtained in the proof of Theorem 4.1 the following bound on $|\widehat{y}_i - y_i^*|$, which applies to *any* forward algorithm:

$$|\widehat{y}_i - y_i^*| \leq \frac{1}{K} + \max\left\{\frac{1}{i - \ell_k + 1}, \frac{1}{r_k - i + 1}\right\}.$$

Figure 1: Lower bounds for forward algorithms

This implies the same bound on $\ell(y_i, \widehat{y}_i)$. Summing over trials as in the proof of Theorem 4.1 gives the claimed result.

Interestingly, note that in order to prove the bound on log-IVAP in the entropic loss setting, we needed a bound on $|\delta_i|$, which works uniformly over all forward algorithms, including the one that always produces the worst possible estimate $y_i'$ being the opposite of the true label $y_i$.

## H   On Heavy-$\gamma$

We first show that the weight $c$ for Heavy-$\gamma$ should not be taken too high.

**Lemma H.1.** *The worst-case excess leave-one-out loss of Heavy-$\gamma$ is $\Omega(c)/t$.*

*Proof.* At position $i$ on the all-zero sequence $\boldsymbol{y} = (0, \ldots, 0)$, Heavy-$\gamma$ predicts with $\widehat{y}_i = \frac{c\gamma}{c+t-i}$ and incurs loss $\widehat{y}_i^2$. Its leave-one-out loss satisfies

$$\frac{1}{t} \sum_{i=1}^{t} \left( \frac{c\gamma}{c+t-i} \right)^2 \geq \frac{1}{t} \int_0^t \left( \frac{c\gamma}{c+t-x} \right)^2 \mathrm{d}x = \frac{c\gamma^2}{c+t},$$

and since the isotonic regression has no loss, this is also the excess leave-one-out loss. Similarly, the leave-one-out loss on the all-one sequence $\boldsymbol{y} = (1, \ldots, 1)$ is at least $\frac{c(1-\gamma)^2}{c+t}$. Taking the worse of these two cases yields the claimed bound. $\qquad\square$

Next we show that the weight $c$ for Heavy-$\gamma$ should not be taken too low.

**Lemma H.2.** *Fix $\alpha \in [0, \frac{1}{3}]$. The worst-case excess leave-one-out loss of Heavy-$\gamma$ with weight $c = t^\alpha$ is at least $t^{-\frac{1+\alpha}{2}}$.*

*Proof.* We split the sequence into $K = t^{\frac{1}{2}(1-\alpha)}$ segments, each of length $n = t^{\frac{1}{2}(1+\alpha)}$ (so that $nK = t$). In each segment, we have an increasing frequency $\frac{k}{K}$, so the adjacent frequencies are separated by $\frac{1}{K} = t^{-\frac{1}{2}(1-\alpha)}$. Now, the learner by adding $c = t^\alpha$ mass can change the frequency in a given segment by an amount:

$$\frac{t^\alpha}{n + t^\alpha} \simeq \frac{t^\alpha}{t^{\frac{1}{2}(1+\alpha)} + t^\alpha} \simeq \frac{t^\alpha}{t^{\frac{1}{2}(1+\alpha)}} = \frac{1}{K},$$

so that the segments are well separated and will not influence each other.

Now note that $c = t^\alpha = n^{\frac{\alpha}{\frac{1}{2}(1+\alpha)}} \leq \sqrt{n}$. Repeating the analysis for excess leave-one-out loss on a single segment with $p = \frac{k}{n}$ gives:

$$k \left( \frac{n - k + c(1 - \gamma)}{n - 1 + c} \right)^2 + (n - k) \left( \frac{k + c\gamma}{n - 1 + c} \right)^2 - np(1-p)$$

$$= \frac{p(1-p)\left(n + 2c + \frac{c^2}{n}\right) + \frac{c^2(p-\gamma)^2}{n}}{(1 - \frac{1}{n} + \frac{c}{n})^2} - np(1-p)$$

$$= 2p(1-p) + O\left( \frac{c^2(p-\gamma)^2}{n} \right),$$

which is constant per segment. Summing over segments and dividing by $t$ gives the excess leave-one-out loss equal to $K/t = t^{-\frac{1}{2}(1+\alpha)}$. Hence $\alpha$ must be at least $\frac{1}{3}$ to have the excess leave-one-out loss no more than $O(t^{-\frac{2}{3}})$.

**Note:**   There is some hand-waving in this argument, because we assume the algorithm predicts with a constant in a given segment. In fact the algorithm can sometimes split the segment into two subsegments. But if we assume all 1's precede all 0's, this can only happen for $O(c)$ points in each interval (as the frequency of the initial part of the segment will then exceed the frequency of the whole segment, and there will be no reason to split the segment anymore). For all these points we

will make a prediction that is at most $O(1/K)$ off from the isotonic regression function. The extra loss incurred is hence of order $O(1/K)$ per point. As there are $K$ intervals with each at most $O(c)$ points, the total extra error is of negligible order $O(c)$. $\square$