[Reviews · NeurIPS 2017]

Reviewer 1



In this paper, the authors study isotonic regression on linear orders for online settings. The authors mention that it was shown that online isotonic regression can not be learned in a fully adversarial model, and this motivates them to tackle the problem by proposing a more practical random permutation model. Some typos and grammatical inconsistencies aside, the paper is well written, clear and addresses an interesting topic. However, it is difficult to confirm the validity and the veracity of the claims formulated in the paper due to the absence of experiments. To support the claims formulated in the paper it would be good to simulate certain examples and to demonstrate that the isotonic regression on linear orders operate in online settings.

Reviewer 2



Summary: The authors study the online Isotonic regression problem under the random permutation model. The adversary picks the set of instances x_1, ..., x_n beforehand, and reveals it according to a random permutation. At round t, the instance x_i_t is revealed, and the algorithm predicts y-hat_i_t. Next, the adversary reveals y_i_t, and the algorithms incurs the squared loss (y-hat_i_t - y_i_t)^2. The goal is to minimize regret. From a previous work, it is known that if the sequence of instances is arbitrary, the regret can be linear in the number of rounds. And the same work also studied the regret under the fixed design model where the sequence of instances is fixed and revealed to the algorithm beforehand, and gave an algorithm to achieve an optimal ~T^(1/3) regret. In this work the authors present results in the significantly more general random permutation model, and present a (computationally inefficient) algorithm achieving optimal ~T^(1/3) regret, and show that a large family of efficient algorithms (Forward Algorithms) achieve ~T^(1/2) regret in the general case, but an optimal ~T^(1/3) regret in the case that revealed labels are isotonic. Opinion: I think the problem of online Isotonic regression is interesting, and the authors present good results it in a very natural and compelling model (Esp given that the completely adversarial model is hopeless). The efficient algorithms presented do not achieve optimal bounds, but I think the authors make a solid contribution in understanding the behavior of a large class of algorithms, with good regret bounds; including conjecturing an efficient optimal algorithm. I think the techniques in this paper will be interesting and necessary for making further progress. I recommend acceptance. One concern: I would like the authors to spend some time explaining why this notion of regret is natural since the algorithm/learner is not constrained to isotonic outputs, it seems to me that the current notion of regret could be negative? Edit: I have read the authors' feedback, and the review stands unchanged

Reviewer 3



I am not an expert in online learning, and did not read the proofs in the appendix. My overall impression of the paper is positive, but I am not able to judge the importance of the results or novelty of the analysis techniques. My somewhat indifferent score is more a reflection of this than the quality of the paper. Summary: The authors study isotonic regression in an online setting, where an adversary initially chooses the dataset but the examples are shown in random order. Regret is measured against the best isotonic function for the data set. The main contribution of the paper seems to be in Section 4, i.e a class of "forward algorithms" which encompass several well known methods, achieve sqrt(T) regret. The authors also prove a bunch of complementary results such as lower bounds and results for different settings and loss functions. Can the authors provide additional motivation for studying the random permutation model? I don't find the practical motivation in lines 46-47 particularly convincing. It might help to elaborate on what is difficult/interesting for the learner this setting and what makes the analysis different from existing work (e.g. [14]). Section 3: How do Theorems 3.1 and 3.2 fit into the story of the paper? Are they simply some complementary results or are they integral to the results in Section 4? - The estimator in (3) doesn't seem computationally feasible. Footnote 2 states that the result holds in expectation if you sample a single data and permutation but this is likely to have high variance. Can you comment on how the variance decreases when you sample multiple data points and multiple permutations? Clarity: Despite my relative inexperience in the field, I was able to follow most of the details in the paper. That said, I felt the presentation was math driven and could be toned down in certain areas, e.g. lines 149-157, 242-250 While there seem to a some gaps in the results, the authors have been fairly thorough in exploring several avenues (e.g. sections 4.3-4.6). The paper makes several interesting contributions that could be useful for this line of research. ---------------------------------------------------- Post rebuttal: I have read the authors' rebuttal and am convinced by their case for the setting. I have upgraded my score.